# Paliperidone Palmitate Every Three Months (PP3M) 2-Year Treatment Compliance, Effectiveness and Satisfaction Compared with Paliperidone Palmitate-Monthly (PP1M) in People with Severe Schizophrenia

**DOI:** 10.3390/jcm10071408

**Published:** 2021-04-01

**Authors:** Juan J. Fernández-Miranda, Silvia Díaz-Fernández, Domenico De Berardis, Francisco López-Muñoz

**Affiliations:** 1Asturian Mental Health Service Área V-Servicio de Salud del Principado de Asturias (SESPA), 33211 Gijón, Spain; juanjofmiranda@gmail.com (J.J.F.-M.); marmotillazz@gmail.com (S.D.-F.); 2Asturian Institute on Health Research (ISPA), 33011 Oviedo, Spain; 3Faculty of Health Sciences, University Camilo José Cela, 28692 Madrid, Spain; flopez@ucjc.edu; 4National Health Service, Department of Mental Health, Psychiatric Service of Diagnosis and Treatment, “G. Mazzini” Hospital, ASL 4, 64100 Teramo, Italy; 5Neuropsychopharmacology Unit, Hospital 12 de Octubre Research Institute (i+12), 28041 Madrid, Spain; 6Portucalense Institute of Neuropsychology and Cognitive and Behavioural Neurosciences (INPP), Portucalense University, 4200-072 Porto, Portugal; 7Thematic Network for Cooperative Health Research (RETICS), Addictive Disorders Network, Health Institute Carlos III, MICINN and FEDER, 28029 Madrid, Spain

**Keywords:** schizophrenia, antipsychotic, treatment effectiveness, treatment satisfaction, treatment compliance, long-acting injectable, paliperidone palmitate every three months

## Abstract

Paliperidone palmitate every three months (PP3M) is expected to facilitate patient’s treatment compliance and satisfaction. The objective here was to compare PP3M treatment compliance and satisfaction, effectiveness and tolerability, with paliperidone palmitate-monthly (PP1M) in patients with severe schizophrenia. A 24-month prospective, open-label study of patients with severe schizophrenia treated with PP3M after at least 2 years of stabilization with PP1M (*n* = 84) was carried out. Treatment satisfaction was assessed with the Treatment Satisfaction Questionnaire for Medication (TSQM) and with a Visual Analogue Scale (VAS). Effectiveness was measured with psychiatric hospital admissions and the Clinical Global Impression-Severity (CGI-S) scale. Tolerability assessments included laboratory tests, weight and adverse effects. Reasons for treatment discontinuation were recorded. CGI-S significantly improved after 24 months. Three patients changed back to PP1M due to adverse effects, and four were hospitalized. There were neither abandoning nor significant changes in weight or biological parameters, and lower incidence of side effects, with PP3M treatment. TSQM and VAS scales increased. No differences were found related to doses. Apart from somewhat improvement in treatment adherence, effectiveness, and tolerability, patients with severe schizophrenia lengthy treated with PP1M showed more satisfaction with PP3M, even those who needed high doses to get clinical stabilization.

## 1. Introduction

In the schizophrenia management, long-acting injectable (LAI) antipsychotics contribute to the successful maintenance of treatment by improving non-adherence and preventing relapses. Although the use of second generation antipsychotics (SGAs) LAIs has been recommended in the last years to treat those patients at high risk of discontinuation [1,2], they can have more advantages like relapse prevention, high tolerability and low mortality [3,4]. Treatment with paliperidone palmitate-monthly (PP1M) has been reported to significantly reduce hospitalizations and that it is generally well tolerated [5,6,7].

Paliperidone palmitate 3-monthly (PP3M) formulation is a long-acting, injectable antipsychotic treatment approved for the maintenance treatment of adult patients with schizophrenia. PP3M formulation is the only available LAI antipsychotic that offers an extended three-month window of stable plasma drug concentration, enabling only four injections per year [8]. Further, it offers advantages apart from long intervals between injections, like tolerability and enhanced patient compliance, and provides significant improvement in psychotic symptoms. This formulation allows patients to maintain therapeutic paliperidone plasma levels with fewer injections, which could subsequently improve functional outcome and quality of life with a sufficient follow-up period [9]. As PP3M requires injections only four times a year and this is expected to facilitate patient’s treatment compliance and satisfaction.

In summary, PP3M is a valuable antipsychotic treatment option in the long-term treatment of schizophrenia, and its utility should not be limited to patients with poor adherence [10]. As current evidence supports the efficacy and tolerability of PP3M compared to paliperidone palmitate monthly (PP1M) and placebo, PP3M appears to be a viable treatment option for patients previously maintained on PP1M. However, to truly establish the place of PP3M in therapy relative to other oral antipsychotics and LAIs in real-world settings, more research is needed [8,9].

Indeed, although clinical trials have demonstrated that PP3M is an efficacious, safe, and tolerable treatment in patients with schizophrenia [11,12,13], the long-term effectiveness of PP3M in studies lasting more than a year has never been studied. Further, there are no specifically researches in severely ill patients. Moreover, there are no studies to know if some patients need higher doses than labeled to get clinical stabilization, without intolerable side effects.

The present study compares, in a real-life setting, and for a long-time period, PP3M treatment compliance and satisfaction, effectiveness and tolerability, to PP1M in patients with severe schizophrenia previously stabilized with PP1M for at least two years. Further, it explores the doses needed to achieve the best outcomes and its tolerability, even if they are higher than those studied in randomized clinical trials (RCTs).

## 2. Methods

A 24-month prospective, observational, open-label study of patients with severe schizophrenia (Clinical Global Impression-Severity [CGI-S] ≥ 5 at the beginning of PP1M treatment) treated with PP3M after at least 2 years of stabilization with PP1M was carried out. Retention in treatment since first injection was measured, and also all-causes for treatment discontinuation. Clinical severity of illness was measured with CGI-S scale. Treatment satisfaction with PP3M vs. PP1M was assessed with the Treatment Satisfaction Questionnaire for Medication (TSQM) and with a visual analogue scale (VAS; ranged from 1, not at all satisfied, to 10, extremely satisfied).

Effectiveness was measured with the number of hospitalizations due to psychiatric decompensation and with the CGI-S. Tolerability assessments included extrapyramidal symptoms and other movement disorders, weight, adverse effects reported and injection-site pain or reaction every three months; and laboratory tests (haematology, biochemistry and prolactin levels) every year. Other psychiatric medications and also all-cause for treatment discontinuation were registered.

The sample (*n* = 84) included all those subjects that met the criteria for diagnosing schizophrenia and severe symptoms and impairment, with a GCI-S scoring of ≥ 5, according to the treating clinicians, and treated with PP3M after at least 2 years of stabilization with PP1M. They were the first ones who were offered to change from PP1M to PP3M just with the aim of reducing injections and as a result to improve comfort and quality of life. Other clinicians besides those treating the patients were part of the study assessing changes in CGI-S and side effects. All patients were undergoing treatment in a case managed program for people with severe mental illness, mainly schizophrenia, with psychosocial and pharmacological integrated approach. Recruitment was made between July 2017 and June 2018, with the follow-up between July 2018 and June 2020.

In this study, non-compliance is defined as the complete discontinuation of PP3M for more than 30 consecutive days. Hospitalisation is defined as the subjects’ psychiatric admission associated with psychotic and/or other psychiatric symptoms. High-dose therapy is defined based on doses exceeding the approved maximum recommended dose (525 mg/3 months).

All patients (or their legal representatives, if appropriate) signed informed consent forms to begin their treatments. The study was carried out in accordance with the Code of Ethics of the World Medical Association-WMA ethical principles (Declaration of Helsinki), and was approved by the Ethical Clinical Research Committee of the Asturian Health Service (Proyecto de Investigacion n° 88/16).

The main statistical analyses (descriptive and inferential) were to compare treatment discontinuation, hospital admissions, scales scoring, side effects and laboratory test results before and after PP3M treatment. Chi^2^ was used for qualitative variables, with the McNemar test specifically used to compare paired proportions. A Student’s t was used for paired data for quantitative variables. The confidence interval was established at 95%. The “R Development Core Team” program (version 3.4.1) and Package MASS (7.3–45 version) was used for data processing.

## 3. Results

Sociodemographic and clinical characteristics of the studied patients are shown in Table 1. There were no significant differences between standard or high doses groups except in age, in living alone or with the own family, and previous hospital admissions, higher in the second group (*p* < 0.05).

CGI-S at baseline was 4.1 (0.5), with significantly improvement after 24 months (*p* < 0.01) (Table 2). Only three patients preferred change back to PP1M due to adverse effects (mainly parkinsonism). There were no voluntary discontinuations with PP3M treatment. Four patients were referred to hospital psychiatric ward due to decompensation (in the previous two years, nine) (Table 2). There were neither significant changes in weight or prolactin levels nor biological parameters alterations, although both decreased. Furthermore, lower incidence of Parkinsonism (treated), sedation and orthostatic hypotension was reported. There was an increase in TSQM (from ‘satisfied’ to ‘very satisfied’; *p* < 0.01) and VAS (from 7.6 (0.9) to 9.1 (0.8); *p* < 0.001) between 1M and 3M PP treatment (Table 2). Reasons reported for higher satisfaction were less injections/year, less sedation and lower feeling of being medicated. No statistical differences between both groups (standard vs. high doses) were found. Adverse effects reported, weight and biological parameters are shown in Table 3.

## 4. Discussion

### 4.1. PP1M and PP3M Efficacy

It is well demonstrated the clinical efficacy of antipsychotic (AP) drugs in the treatment of schizophrenia, and in especial regarding positive symptoms. This efficacy correlates with their ability to act in specific dopamine and serotonin systems receptors, and explains why second generation APs are being increasingly used, whereas the use of conventional or first generation ones is decreasing [14]. Long-acting injectable (LAI) antipsychotics were developed to improve treatment compliance and secondarily patient outcomes. Nevertheless, effectiveness, further than efficacy in randomized clinical trials (RCTs), and tolerability of antipsychotics is important to increase treatment compliance, and consequently to improve clinical and rehabilitation treatment outcomes in people with schizophrenia in routine practice [4,15]. The superiority of LAI-APs over OAPs in effectiveness is more evident in mirror studies [15] and in cohort studies [16] than in RTCs, influenced by several biases. This has been demonstrated in naturalistic studies, even in those with patients on high doses of LAI-SGAs [17].

PP1M treatment has proven in RTCs clinically significant improvement in the patients with schizophrenia, reducing hospitalizations [5,11,12,13,18,19]. Furthermore, it improves social performance and functionality, and proves to be safe and well tolerated, in naturalistic studies [6,20]. PP three-month injection is an atypical antipsychotic containing a racemic mixture of the active ingredient paliperidone, and utilizes nanocrystal technology similar to the PP1M, but with increased particle size, allowing for an extended sustained release [21].

Several studies designed to evaluate the efficacy and safety of the three-month formulation of paliperidone palmitate vs placebo in delaying time to relapse of schizophrenia symptoms in patients previously treated with PP1M for at least four months have been carried out [21,22,23]. Overall, safety and tolerability were similar to the one-month formulation, and the slower profile of PP3M supported a dosing interval of three monthly administrations in patients with schizophrenia [22]. The same was found in studies on non-inferiority compared to PP1M [23]. A randomized, multicenter trial conducted in eight countries, showed that, compared with placebo, PP3M significantly delayed time to relapse in patients with schizophrenia, it was generally tolerable and has a safety profile [9].

### 4.2. PP3M Effectiveness

The symptomatic and functional outcomes after treatment with paliperidone palmitate three-month formulation for 52 weeks in patients with clinically stable schizophrenia have been studied [24], evaluating the efficacy and safety of converting patients with schizophrenia stabilized with PP1M to PP3M in a naturalistic clinical setting. Its results were similar to those observed in previous randomized clinical trials of PP3M, and underline the importance of continuous maintenance treatment.

Another research [25] identified patient and disease characteristics during PP1M treatment associated with greater likelihood of achieving remission after transition to PP3M. Patients with early clinically meaningful improvements in disease symptoms and severity while establishing stable PP1M dosage were more likely to achieve remission after transition to PP3M. A “real-world” study showed clinical effectiveness of PP3M in early psychosis patients [26]. Another one [10] summarized available evidence for PP3M that can be interpreted in clinical practice, as low number-needed-to-treat for relapse prevention (six-month estimate: 4.8; 12-month estimate: 3.4). A pragmatic clinical study assessed goal attainment among patients on PP3M and whether patients achieved symptomatic remission at the study endpoint. The results indicated that continued treatment with PP3M may facilitate patients’ personal goals and reduce disability [27]. However, none of these studies are longer than a year, contrary to the present one. Further, the need of confirming PP3M outcomes through long-term maintained treatment is clear, taking into account that people with schizophrenia is treated for many years, if not long-life.

Regarding to hospital admissions, RCTs comparing oral (OAP) and LAI APs have often failed to show any clear advantages of LAI-APs over OAPs, and its superiority what concerns to relapse prevention is supported by naturalistic studies [4,28]. Our naturalistic research confirms the PP3M maintained effectiveness in preventing patient relapses and in reducing hospitalizations and clinical severity: the CGI-S score of patients receiving PP3M treatment for two years is lower than while treating with PP1M. This lower illness severity is probably related to the reduction in hospital admissions reached with PP3M treatment.

It is well known that medication nonadherence and related relapses increase the disease illness associated with schizophrenia, in especial in those patients with severe symptoms and impairment [29,30]. To improve treatment compliance, SGA-LAIs has been recommended [2,3]. The better performance of LAIs over OAPs for treatment retention is clearer in naturalistic studies than in RTCs [1,4,16], and needs to be confirmed in studies lasting more than a year. In a research comparing adherence and costs pre- and post-transition to PP3M [31], authors found that transitioning to PP3M was associated with an improvement in adherence and suggest PP3M may enhance adherence while remaining cost neutral. Our findings show a high level of long-term treatment compliance with PP3M in patients with a high rates of treatments abandoning, what in all probability allowed achieve the clinical goals. Once again, dosage did not influence this outcome.

Summing up, PP3M maintenance showed in our research to be remarkable effective in severely ill patients with schizophrenia, although some of them needed higher doses than 525 mg/3M to achieve clinical stabilization. These results are linked to the fact of patients were markedly adherents for a long time to this AP formulation.

### 4.3. Tolerability and Treatment Satisfaction

The findings for PP3M safety were consistent with those seen in other clinical trials with PP1M [22]. The most common adverse reactions with an incidence of 5% or more were injection-site reaction, weight increase, headache, upper respiratory tract infection, akathisia, and parkinsonism [21]. Discontinuation of treatment with PP3M due to adverse effects in one of the clinical trials was 5.1% during the open-label phase, but no subjects discontinued PP3M due to adverse effects during the double-blind phase [9]. In a evaluation of the safety of PP3M vs. PP1M treatment with regard to extrapyramidal symptoms in patients with schizophrenia previously stabilized on PP1M treatment [10], both formulations exhibited comparable incidence. Moreover, the authors reviewed PP3M tolerability with data as high number-needed-to-harm (akathisia, 27.1; tremor, 80.0; dyskinesia, −132.6; parkinsonism, 160.0). They confirmed the relative benefits and low propensity for adverse events with PP3M.

An investigation about injection site reactions and pain following PP1M and PP3M administration [32], PP1M and PP3M injections were well tolerated. Incidence of induration, redness, and swelling were low and mostly mild in both treatment groups, and injection site reactions and pain were low and similar between them.

In the present study, we have found a lower incidence of side effects reported and of movement disorders. Further, only three patients preferred change back to PP1M due to adverse effects. There were neither significant changes in weight or prolactin levels nor biological parameters alterations, although in general all biological parameters decreased. There was also a significant diminish in sedation, highlighted by the patients as one of the main reasons for their satisfaction with PP3M therapy. Our research confirms all that knowledge provided by RTCs and naturalistic studies about the safety and tolerability of PP3M.

Although the data available do not provide conclusive evidence to support the safety of high-dose treatment strategies [17], this study shown the same tolerability in those patients on doses over 525 mg/3M. Nevertheless, previous naturalistic studies with high doses of long-acting risperidone and PP1M have shown rather good tolerability and adherence [33,34]. This is a remarkable fact, because allow clinicians to consider higher doses of PP3M to improve its effectiveness with low risk of adverse effects.

Finally, it is remarkable that there was a significant increase in the satisfaction with PP3M therapy, up to be very satisfied, as shown by TSQM and VAS. We have to consider that previous satisfaction with PP1M was high already. Reasons reported for higher satisfaction were less injections per year and lower feeling of being medicated, apart from less side effects, mainly sedation. Further, what is important, no differences were found between those on standard or on high doses.

### 4.4. Study Limitations and Strengths

Although this study was designed to compare PP1M to PP3M in regular clinical settings, its limitations include the risk of confounding factors due to non-randomization: it is an open-label, non-randomised study under pragmatic conditions, and there is no control group, which may mean lower internal validity. We have implemented a study using the same patients with previous two years treated with PP1M as a comparator. Other possible limitations are the relatively small sample, that no formal side effects assessment scales have been applied, and that we have used the CGI-S as an acceptable measure of a change in severity, although it is a non-specific instrument.

The main strength of our study is that assesses treatment adherence and satisfaction, tolerability and effective doses of PP3M in a real-world sample of severely ill people, and for a long period (24 months). That allows us to measure retention in treatment of people with severe schizophrenia, one of the main problems with them, and to link it to satisfaction with this pharmacological therapy. Further, we need to also highlight the need of some patients for doses over the standard ones, in order to achieve similar outcomes without increasing side effects.

## 5. Conclusions

Apart from somewhat better treatment adherence, effectiveness (lower severity of illness and fewer hospitalizations) and tolerability, patients with severe schizophrenia lengthy treated with PP1M showed more satisfaction with PP3M. These outcomes were no different in those patients who needed high doses to get clinical stabilization. Therefore, this formulation allows patients not only to improve treatment outcomes but also to feel more satisfied with it. Further, allowing patients and caregivers more time to focus on other aspects of treatment, such as psychological and social interventions, provided in the integrated case managed program where they have been treated.

Although this study rated all the patients as severely ill according to the CGI-S, and the results presented here may not generalize to not severely ill populations, they may be useful to guide clinicians to manage complex patients, as many are in routine practice.

Nevertheless, we think that long-term naturalistic studies with larger sample sizes on specific real-world populations are required to confirm our results

## Figures and Tables

**Table 1 jcm-10-01408-t001:** Demographic and previous clinical data.

	Total *n* = 84	Standard Doses *n* = 50	High Doses *n* = 34
Gender: male (*n*)	44	26	18
Age [Av(SD)] years *	42.1 (7.6)	39.7 (6.6)	43.2 (7.5)
Working	32	19	13
Living alone/own family *	34	22	12
Previous tt. duration [Av(SD)] years	16.7 (8.2)	15.9 (8.2)	17.1 (7.7)
Previous tt. discontinuation (*n*)	40	24	16
Previous hospital admissions (*n*) *	35	20	15

(*n*, %): number, percentage; Av (SD): average (standard deviation); tt.: treatment; *: *p* < 0.05.

**Table 2 jcm-10-01408-t002:** Clinical Global Impression-Severity (GCI), Visual Analogue Scale (VAS) and Treatment Satisfaction Questionnaire for Medication (TSMQ) scores, and number of hospital admissions.

*n* = 84	GCI-S *	VAS **	TSMQ *	Hospital *
PP 1-month	4.1 (0.5)	7.8 (0.9)	75.2 (12.6)	9
PP 3-month	3.4 (0.5)	9.1 (0.8)	85.6 (11.9)	4

CGI-S: Clinical Global Impression-Severity; TSQM: Treatment Satisfaction Questionnaire for Medication; VAS: Visual Analogue Scale; Hospital: hospitalizations; *: *p* < 0.01; **: *p* < 0.001.

**Table 3 jcm-10-01408-t003:** Side effects, laboratory tests and weight.

*n* = 84	Baseline	2 Years
Parkinsonism (treated) *	15	11
Sedation **	16	8
Anticholinergic effects	9	9
Hypotension *	12	8
Hyperprolactinaemia	13	12
Hyperglycaemia	14	14
Lipid alteration	20	17
Blood parameters alteration	12	13
Hepatic function altered	11	11
Weight [Av(SD)] Kg.	83.6 (11.3)	81.2 (9.9)

Av (SD): average (standard deviation); *: *p* < 0.05; **: *p* < 0.01.

## Data Availability

Data are available upon request.

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
