# Peer review of "Paliperidone Palmitate Every Three Months (PP3M) 2-Year Treatment Compliance, Effectiveness and Satisfaction Compared with Paliperidone Palmitate-Monthly (PP1M) in People with Severe Schizophrenia"

_jcm, 2021, doi:10.3390/jcm10071408_

Round 1
Reviewer 1 Report
This is a very important paper in terms of clinical results. My only suggestion is that there is no information concerning the proposed mechanism of PP3M which takes away from the interest of the paper to non-clinicians. I this regard I recommend the authors considering referencing the work related to schizophrenia as well (Blum et al., Medical hypothesis 2014 82 (5) 606-614
Also please use p values with P < 0.05 etc it is missing in a few cases and also make sure that all abbreviatons are spelled out.
Author Response
Dear J Clin Med editors,
Thank you for the reviewers comments. Enclosed you`ll find the changes made, point by point (in red color in the text):
Reviewer 1
My only suggestion is that there is no information concerning the proposed mechanism of PP3M which takes away from the interest of the paper to non-clinicians. I this regard I recommend the authors considering referencing the work related to schizophrenia as well (Blum et al., Medical hypothesis 2014 82 (5) 606-614
Made a short comment in Discussion and enclosed the reference.
Also please use p values with P < 0.05 etc it is missing in a few cases and also make sure that all abbreviatons are spelled out.
P values and abbreviations reviewed
Reviewer 2 Report
This is an interesting study, however there are some points which need revision:
Methods and results: Are the authors able to present some additional information - e.g. PANSS scale results, data about general functioning of the patients, e. g. education, occupation etc.
Results: The authors should present statistical data (e.g. p value) about side effects etc (Table 3).
Author Response
Dear J Clin Med editors,
Thank you for the reviewers comments. Enclosed you`ll find the changes made, point by point (in red color in the text):
Reviewer 2
Methods and results: Are the authors able to present some additional information - e.g. PANSS scale results, data about general functioning of the patients, e. g. education, occupation etc.
Additional patients functional information included in Table 1.
Results: The authors should present statistical data (e.g. p value) about side effects etc (Table 3).
P value included in Table 3
PLEASE NOTE: in red in the text:
Round 2
Reviewer 2 Report
The authors have corrected the manuscript according to my comments